# A humanized IFN-γ mouse model reveals skin eschar formation, enhanced susceptibility and scrub typhus pathogenesis

Ryan H. Cho[1], Lihai Gao[1], Hui Wang[2], Yixuan Zhou[3], Casey Gonzales[2], Dario Villacreses[1], Emmett A. Dews[1], Xiaofei Zhou[4], Ruili Lv[4], Hema P. Narra[2], Lynn Soong[1,2,5,6]*, Yuejin Liang[1,5,6]*

1 Department of Microbiology and Immunology, University of Texas Medical Branch, Galveston, Texas, United States of America, 2 Department of Pathology, University of Texas Medical Branch, Galveston, Texas, United States of America, 3 Department of Neurology, University of Texas Medical Branch, Galveston, Texas, United States of America, 4 Biocytogen Pharmaceuticals (Beijing) Co., Ltd., Beijing, China, 5 Institute for Human Infections and Immunity, University of Texas Medical Branch, Galveston, Texas, United States of America, 6 Center for Tropical Diseases, University of Texas Medical Branch, Galveston, Texas, United States of America

* lysoong@utmb.edu (LS); yu2liang@utmb.edu (YL)

## Abstract

Scrub typhus, caused by *Orientia tsutsugamushi* (*Ot*) bacteria, is a serious acute febrile illness associated with significant mortality. No effective vaccine is currently available, largely due to the complex *Ot* strain diversity and an incomplete understanding of protective immune mechanisms. To overcome these challenges, there is a critical need for a suitable animal model that mimics human disease through the natural route of infection. Here, we report for the first time that a genetically engineered humanized mouse strain (with triple knockout/knock-in of IFN-γ and its receptors, abbreviated as hIFNG/hIFNGR), exhibits increased susceptibility to intradermal *Ot* infection compared to wild-type (WT) mice. This is evidenced by greater body weight loss, elevated bacterial burden, and reduced expression of interferon-stimulated genes (ISGs). hIFNG/hIFNGR mice exhibit pronounced biochemical abnormalities and tissue pathology accompanied by dysregulated T cell and neutrophil responses following infection. Notably, this novel mouse strain with human IFN-γ signaling can develop skin eschar-like lesions resembling those observed in human patients. Overall, our study introduces a promising mouse model to dissect the immunopathogenesis of scrub typhus and evaluate future vaccine candidates.

## Author summary

Scrub typhus is a serious disease caused by the obligately intracellular bacterium *Ot* that spreads to humans through the bite of larval mites called chiggers. Although it is treatable with antibiotics, delayed diagnosis and treatment can

**Data availability statement:** All raw data have been published on the repository Mendeley and are available online: https://data.mendeley.com/datasets/knjp6gypkd/2.

**Funding:** This work was supported partially by the National Institute of Allergy and Infectious Diseases grants (AI132674 to LS and YL; AI156536 to LS; AI179997 to HPN and LS, AI184781 to YL. https://www.niaid.nih.gov/), a UTMB IHII NTT Startup grant (to YL. https://www.utmb.edu/ihii), and a UTMB IHII Pilot grant (to HW. https://www.utmb.edu/ihii). The funders had no role in study design, data collection and analysis, decision to publish, or preparation of the manuscript.

**Competing interests:** The authors have declared that no competing interests exist.

lead to severe infection and fatal outcomes. Unfortunately, we still lack a clear understanding of how this infection causes disease, in part due to the lack of laboratory models that accurately reflect human responses to infection. Our recent reports have suggested an important role of IFN-γ in host protection against *Ot* infection. In this study, we used a new genetically modified mouse strain that carries functionally attenuated human IFN-γ signaling in place of its mouse counterpart. We found that hIFNG/hIFNGR mice are more vulnerable to infection, develop skin eschar lesions like those in patients, and show signs of systemic inflammation and organ damage. Their immune response also resembled what has been observed in human patients. This new mouse model can help scientists better understand the mechanisms as to how this bacterial species causes severe disease outcomes in patients. Once those mechanisms are understood, this mouse model will serve a further purpose as a tool for testing new vaccines and treatments to aid humans at-risk for scrub typhus.

## Introduction

*Orientia tsutsugamushi* (*Ot*) is an obligately intracellular bacterium that causes scrub typhus in humans. It is estimated that the overall pooled seroprevalences of scrub typhus were 10.73% and 22.58% in healthy people and in febrile patients, respectively, mostly within the "tsutsugamushi triangle", which extends from Pakistan in the west to far eastern Russia in the east to northern Australia in the south [1]. Robust epidemiological studies remain limited because only a small number of countries designate scrub typhus as a notifiable disease within their national surveillance systems [1]. *Ot* is primarily transmitted to humans through the bite of infected chigger mites of the *Leptotrombidium* genus, such as *Leptrotrombidium palpale,* a reservoir which maintains the bacterium through transovarial transmission [2]. The clinical manifestations of scrub typhus include fever, headache, myalgia, vomiting, lymphadenopathy, and the characteristic eschar at the site of the chigger bite [3–5]. In severe cases, bacterial dissemination to multiple organs can result in life-threatening complications such as respiratory failure, disseminated intravascular coagulation, meningitis, and encephalitis [6–8]. Although traditionally endemic to countries within the tsutsugamushi triangle, scrub typhus cases have been reported in Africa, South America, and the Middle East [9]. Notably, a recent discovery of *Ot* DNA in chiggers in North Carolina raises concerns about potential emergence in the United States [10]. Currently, there is no licensed vaccine for scrub typhus, partly due to a limited understanding of host-pathogen interactions and the extensive antigenic diversity among *Ot* strains [11,12]. This lack of immunological insight leaves us ill-prepared to combat this neglected mite-borne disease, underscoring the urgent need to elucidate the mechanisms underlying *Ot* pathogenesis.

 Animal models that closely recapitulate human scrub typhus are essential for advancing immunological studies and evaluating vaccine candidates. Reported animal models include intraperitoneal, intravenous, and intradermal routes of infection

in mice, as well as intradermal and live chigger challenge models in non-human primates [13–21]. The intraperitoneal and intravenous inoculation models, which involve the direct introduction of bacteria into the peritoneal cavity or blood-stream, respectively, allow for systemic dissemination of the bacteria and mimic severe scrub typhus and lethal outcomes [20,22,23]. However, these routes bypass the natural mode of transmission, in which bacteria are introduced by the bite of an infected chigger mite and disseminate via the lymphatic system before bloodstream invasion [21,24]. The intradermal or subcutaneous mouse models, which better mimic the natural route of Ot infection, are commonly used to study bacterial dissemination and pathogenesis [14,17,18,24–27]. These models, though, are not effective for modeling severe scrub typhus disease. While intradermal infection leads to systemic bacterial dissemination and elicits host protective immune responses, immunocompetent mice (inbred C57BL/6 or outbred CD-1 mice, etc.) are largely resistant and develop only mild or no major disease manifestations [14,18,27]. Thus, developing a susceptible and immunocompetent mouse model is critical for investigating bacterial dissemination in humans, elucidating protective host immunity, and evaluating vaccine candidates against human relevant disease states.

It is known that Ot infection elicits strong type 1 immunity in both patients and animal models [28–31]. We recently demonstrated that mice deficient in IFN-γ receptor signaling (Ifngr1$^{-/-}$) are highly susceptible to intradermal Ot infection, exhibiting 100% lethality even at very low infectious doses [24]. In addition, these deficient mice display eschar-like lesions that are not observed in wild-type (WT) mice [24], highlighting the role of IFN-γ signaling in host protection against Ot infection. Although Ifngr1$^{-/-}$ mice are highly susceptible to Ot infection, the deficiency of IFN-γ signaling limits their ability to model the complex and coordinated immune responses observed during natural infections. As a result, they are not suitable for comprehensive studies of immunopathogenesis, vaccine development, or host protective immunity. Given that humans with intact IFN-γ signaling remain susceptible to Ot, we hypothesize that human susceptibility may stem from comparatively attenuated IFN-γ signaling than that observed in mice. Although IFN-γ signaling is functionally conserved between humans and mice, species-specific differences may lead to differential regulation of downstream genes and contribute to the observed differences in susceptibility to infection [32,33].

In this study, we establish a novel mouse model of scrub typhus by using a genetically engineered humanized mouse strain hIFNG/hIFNGR that lacks murine IFN-γ signaling but is reconstituted with human IFN-γ signaling. This mouse model exhibited increased susceptibility to Ot infection compared to WT mice, as demonstrated by greater body weight loss, higher clinical disease scores, and elevated bacterial burdens in multiple organs. Further analysis demonstrated the impaired interferon-stimulated gene expression and dysregulated innate and adaptive immune cell responses in infected hIFNG/hIF-NGR mice. Notably, intradermal Ot infection in hIFNG/hIFNGR mice resulted in eschar-like lesions that closely resemble those observed in human patients. Together, this novel mouse model, combined with intradermal infection, represents a promising platform for future scrub typhus studies on bacterial dissemination, immunopathogenesis, and vaccine development.

## Materials and methods

### Ethics statement

UTMB complies with the USDA Animal Welfare Act (Public Law 89–544), Health Research Extension Act of 1985 (Public Law 99–158), the Public Health Service Policy on Humane Care and Use of Laboratory Animals, and NAS Guide for the Care and Use of Laboratory Animals (ISBN-13). UTMB is registered as a Research Facility under the Animal Welfare Act and has current assurance with the Office of Laboratory Animal Welfare, in compliance with NIH policy. Infections were performed following Institutional Animal Care and Use Committee approved protocols (2101001A) at the UTMB in Galveston, TX.

### Generation of humanized hIFNG/hIFNGR mouse strain

The exons 1–4 of mouse Ifng gene that encode the whole molecule (ATG to STOP codon) were replaced by human counterparts in B-hIFNG mice. The promoter, 5'UTR, and 3'UTR region of the mouse gene were also retained. FRT-flanked Neo resistance positive selection cassette was inserted downstream of the mouse Ifng 3'UTR. Homologous regions,

spanning 4.5 kb upstream of the ATG start codon and 2.7 kb downstream of the 3'UTR, were subcloned from the BAC clone RP23-138P22. Part of exons 2 to exon 6 of mouse Ifngr2 gene that encode the extracellular coding region were replaced by human counterparts in B-hIFNGR2 mice (#110803, Biocytogen). The genomic region of mouse Ifngr2 gene that encodes signal peptide and cytoplasmic portion is retained. FRT-flanked Neo resistance positive selection cassette was inserted intron 4 of the mouse Ifngr2. Homologous regions, spanning 3.9 kb upstream of the exon2 and 4.2 kb downstream of exon 6, were subcloned from the BAC clone RP23-207P1. The exon 1 to part of exon5 of mouse Ifngr1 gene that encode the extracellular coding region were replaced by human counterparts in B-hIFNGR1 mice (#110804, Biocytogen). The genomic region of mouse Ifngr1 gene that encodes cytoplasmic portion is retained. FRT-flanked Neo resistance positive selection cassette was inserted intron 6 of the mouse Ifngr1. Homologous regions, spanning 4.3 kb upstream of the ATG start codon and 4.3 kb downstream of intron 6, were subcloned from the BAC clone RP23-122M4. All the BAC clones (RP23-138P22, RP23-207P1 and RP23-122M4) are derived from the C57BL/6J mouse genomic BAC library. The complete sequence of the three targeting vectors was verified by full-length sequencing. Following linearization, the targeting vector was introduced into C57BL/6J embryonic stem cells via electroporation. Clones were screened and identified by Southern blotting using 5'-probe, 3'-probe, and 3'-Neo probe. Positive clones were injected into BALB/c blastocysts, which were subsequently implanted into pseudo pregnant females. Chimeric male mice were crossed with C57BL/6J females to generate F1 offspring carrying the recombinant allele, which contains the human coding region and the Neo selection cassette. The resulting pups were studied for germinal line transmission of the recombination event by using the PCR strategy. B-hIFNGR1/hIFNG mice (#120819, Biocytogen) were obtained by cross breeding B-hIFNGR1 mice with B-hIFNG mice. B-hIFNGR1/hIFNG/hIFNGR2 (#130949, C57BL/6 background, abbreviated as hIFNG/hIFNGR) mice were obtained by cross breeding B-hIFNGR1/hIFNG mice with B-hIFNGR2 mice. All these experiments are performed by Biocytogen (Waltham, MA) (S1 Fig).

## Analysis of p-STAT1 in immune cells of hIFNG/hIFNGR mice by phosFlow

Mouse splenocytes ($1 \times 10^6$ per well) were isolated from C57BL/6j and hIFNG/hIFNGR mice, starved for 2 hours at 37°C, and then stimulated with various concentrations of mouse or human IFN-γ for 30 minutes (min). Cells were harvested by centrifugation and incubated with blocking and anti-devitalization solutions for 15 min. BV-510 anti-CD3ε (145-2C11) and FITC anti-mCD19 (6D5) were then added for 30 min. Both recombinant cytokine and antibodies were purchased from Biolegend. For STAT1 staining, cells were fixed and permeabilized using BD PhosflowLyse/Fix Buffer 5× and BD Phos-flowPerm Buffer III, and then stained with Phospho-Stat1 (Tyr701) (58D6) Rabbit mAb (#9167S, Cell Signaling Technology, Danvers, MA) for 30 min. Cell samples were acquired for flow cytometry analysis. This experiment is performed by Biocytogen (Waltham, MA).

## Animals, infection, and treatment

Male WT C57BL/6 (#000664) and *Ifngr1*$^{-/-}$ (#003288) mice were obtained from the Jackson Laboratory. hIFNG/hIFNGR mice (#130949, C57BL/6 background) were purchased from Biocytogen (Waltham, MA). All mice were maintained under specific pathogen free conditions and challenged at 11–12 weeks of age. All mouse infection studies were performed in the ABSL3 facility in the Galveston National Laboratory located at UTMB; all tissue processing and analysis procedures were performed in the BSL3 or BSL2 facilities. All procedures were approved by the Institutional Biosafety Committee, in accordance with Guidelines for Biosafety in Microbiological and Biomedical Laboratories. For infection, mice were anesthetized using a VetFlo isoflurane vaporizer in an induction chamber. After anesthesia, the right flank was shaved using an electric trimmer to prepare the injection site. A 20 μL suspension of the *Ot* Karp strain ($3 \times 10^3$ FFU in SPG buffer) was injected intradermally into the right flank using a 0.3 mL insulin syringe with a 31G needle. Mice were monitored for approximately 5 min post-injection to ensure full recovery from anesthesia. Mice were monitored daily for body weight, skin eschar development and disease severity. At 14 days post-infection (dpi), mice were euthanized by $CO_2$ inhalation,

and tissues and blood were collected for further analysis. The disease severity score (ranged from 0–5) was based on an approved animal sickness protocol. The criteria included mobility/lethargy, hunching, fur ruffling, bilateral conjunctivitis, and weight loss: 0- normal behavior; 1- active, some weight loss (< 5%); 2- weight loss (6–10%), some ruffled fur (between shoulders); 3- weight loss (11–19%), pronounced ruffled fur, hunched posture, erythema, signs of reduced food/water intake; 4- weight loss (20–25%), decreased activity, bilateral conjunctivitis, incapable of reaching food/water; 5- non-responsive (or weight loss of greater than 25%) and animal needed to be humanely euthanized.

### Bacterial stock preparation

Bacteria were inoculated onto L929 murine fibroblast monolayers in T150 cell culture flasks and gently rocked for two hours (h) at 37°C. After 2 h, Minimum Essential Medium with 10% fetal bovine serum, 100 units/mL of penicillin and 100 μg/mL of streptomycin were added. Cells were harvested by scraping at 7 dpi, re-suspended in Minimum Essential Medium, and lysed using 0.5 mm glass beads and vortexing for 1 min. The cell suspension was collected and centrifuged at 300 × g for 10 min to pellet cell debris and glass beads. The supernatant from one T150 flask was further inoculated onto new monolayers of five T150 flasks. This process was repeated for a total of six passages. Cells from five flasks were pooled in a 50 mL conical tube with 20 mL medium and 5 mL glass beads. The conical tubes were gently vortexed at 10 second intervals for 1 min to release the intracellular bacteria and placed on ice. The tubes were then centrifuged at 300 × g for 10 min to pellet cell debris, and the supernatant was collected in Oakridge high speed centrifugation bottles, followed by centrifugation at 22,000 × g for 45 min at 4°C to harvest bacteria. Sucrose-phosphate-glutamate buffer (0.218 M sucrose, 3.8 mM $KH_2PO_4$, 7.2 mM $KH_2PO_4$, 4.9 mM monosodium L-glutamic acid, pH 7.0) was used for preparing bacterial stocks, which were stored at -80°C [23, 34, 35]. The same lot of stocks were used for all experiments described in this study. The titers of the bacterial stocks were measured by focus forming assays, as previously described [20].

### qPCR for measuring bacterial burdens

Bacterial burdens were calculated from collected mouse tissues and cultured cells. The samples were initially incubated with proteinase K and lysis buffer at 56°C overnight. DNA was extracted using the DNeasy Blood and Tissue Kit (Qiagen, Germantown, MD) according to the instructions and used for qPCR assays as previously described [20,25,36,37]. The 47-kDa gene was amplified using the primer pair OtsuF630 (5'-AACTGATTTTATTCAAACTAATGCTGCT-3') and OtsuR747 (5'-TATGCCTGAGTAAGATACGTGAATGGAATT-3') primers (IDT, Coralville, IA) and detected with the probe OtsuPr665 (5'-6FAM-TGGGTAGCTTTGGTGGACCGATGTTTAATCT-TAMRA) (IDT, Coralville, IA) by SsoAdvanced Universal Probes Supermix (Bio-Rad, Hercules, CA). Bacterial burdens were normalized to total microgram (μg) of DNA per μL for the same samples. Absolute quantification was performed using a 10-fold serial dilution of the *Ot* Karp 47-kDa plasmid carrying a single copy of the target gene.

### Quantitative reverse transcriptase PCR (qRT-PCR)

RNA was extracted from mouse lungs and brain using the RNeasy Mini Kit (Qiagen, Germantown, MD) according to the manufacturer's instruction, followed by the quality and quantity assessment using a BioTek microplate reader. cDNA was synthesized by using the iScript Reverse Transcription kit (Bio-Rad, Hercules, CA) with the same amount of RNA (1 μg). qRT-PCR was performed using 5 μL of iTaq SYBR Green Supermix (Bio-Rad, Hercules, CA), 1 μL of a forward and reverse primer mix (0.5 μM final concentration of each), and 4 μL of diluted cDNA. Samples were denatured for 30 s at 95°C, followed by 40 cycles of 15 s at 95°C, and 60 s at 60°C, utilizing a CFX96 Touch real-time PCR detection system (Bio-Rad, Hercules, CA). Relative quantitation of transcript levels was calculated using the $2^{-\Delta\Delta Ct}$ method and normalized to glyceraldehyde-4-phosphate dehydrogenase (*Gapdh*). All primers were generated by Integrated DNA Technologies (IDT, Coralville, IA) and are listed in S1 Table. The primer sequences were obtained from PrimerBank (https://pga.mgh.harvard.edu/primerbank).

## Bio-Plex Assay

Whole blood was collected from euthanized mice at 14 dpi, and serum was isolated by using serum separator tubes (BD Bioscience, San Diego, CA). Bacterial inactivation was performed on the samples, as described in our previous study [20]. The Bio-Plex Pro Mouse Cytokine Th1 Panel (Bio-Rad, Hercules, CA) was used to measure serum cytokine levels. The Bio-Rad Bio-Plex Plate Washer and Bio-Plex 200 machine located in the UTMB Flow Cytometry and Cell Sorting Core Lab were used for sample processing and analysis.

## Histology

Tissues were fixed in 10% neutral buffered formalin and embedded in paraffin at the UTMB Research Histology Service Core. Tissue sections (5-μm thickness) were stained with hematoxylin and eosin and mounted on slides. Sections were imaged under an Olympus BX53 microscope, and at least five random fields for each section were captured.

## Clinical pathology

Animal blood chemistry analysis was performed by using the VetScan Chemistry Analyzer (Zoetis, Parsippany-Troy Hills, NJ), according to the manufacturer's instruction. Briefly, mouse serum (100 μL) was uploaded into the VetScan Comprehensive Diagnostic Profile reagent rotor, which was used for quantification of alanine aminotransferase (ALT), albumin (ALB), alkaline phosphatase (ALP), amylase (AMY) total calcium ($CA^{++}$), globulin (GLOB), glucose (GLU), phosphorus (PHOS), potassium ($K^+$), sodium ($NA^+$), total protein (TP), and urea nitrogen (BUN). This analysis was performed at the UTMB ABSL3 animal facility.

## ELISA

Mouse serum samples were collected and human IFN-γ concentrations were analyzed using the Legend Max Human IFN-γ ELISA Kit (Biolegend, San Diego, California) according to the manufacturer's instructions provided by the vendor. To test total IgM and IgG titers in the serum, 96-well plates were precoated with 2 μg/mL of recombinant *Ot* Karp strain TSA56 protein in PBS and blocked with 0.5% BSA. The recombinant TSA56 protein was generated by Genscript (Piscataway, NJ). Serum was diluted 1:3 until endpoint titers were determined. Goat anti-mouse IgM-HRP (#1021-05) and goat anti-mouse IgG-HRP (#1030-05, Southern Biotech, Birmingham, AL) were used at a 1:3000 dilution for detection. 1-Step Ultra TMB ELISA Substrate Solution (Thermo Fisher Scientific, Waltham, MA) was used as the visualizing reagent. Optical density was measured using a Bio-Tek microplate reader. Area under the curve analysis was performed on each curve at every timepoint.

## Flow cytometry

Spleen single-cell suspensions were prepared by passing spleen tissues through 70-μm cell strainers. Red blood cells were removed by using Red Cell Lysis Buffer (Sigma-Aldrich, St. Louis, MO) for 5 min at RT. For surface marker analysis, leukocytes were stained with the Fixable Viability Dye (eFluor 506, Thermo Fisher Scientific, Waltham, MA) for live/dead cell discrimination, blocked with FcγR blocker, and incubated with fluorochrome-labeled antibodies for 30 min. The fluorochrome-labeled antibodies were purchased from Thermo Fisher Scientific and Biolegend as below: Alexa Fluor 700 anti-CD11b (M1/70), APC anti-Ly6G (1A8), PE/Dazzle-594 anti-Ly6C (HK1.4), Percp-Cy5.5 anti-CD11c (N418), FITC anti-MHCII (M5/114.15.2), PE-Cy7 anti-CD3ε (145-2C11), Percp-cy5.5 anti-CD4 (GK1.5), APC-Cy7-anti-CD8a (53–6.7), BV711 anti-CD44 (IM7), APC anti-CD62L (MEL-14), PE CF594-anti-NK1.1 (PK136), FITC anti-CD69 (H1.2F3) and Percp-Cy5.5 CTLA4 (UC10-4B9). BV421 anti-F4/80 (T45-2342) antibody was purchased from BD Bioscience (San Diago, CA). For Foxp3 staining, cells were fixed and permeabilized using Foxp3/ Transcription Factor Fixation/Permeabilization kit, followed by the staining with PE anti-Foxp3 (FJK-16s) for 45 min. Cell samples were fixed in 2% paraformaldehyde

overnight at 4°C, acquired by a BD Symphony A5 and analyzed via FlowJo software version 10 (BD, Franklin Lakes, NJ). The gating strategy of immune cell subsets is based on recent publications [24,36,38].

## Western blot

Lung tissues were homogenized in tissue protein extraction buffer supplemented with 1×protease and phosphatase inhibitor cocktail (Cell Signaling Technology, Danvers, MA). Protein concentrations were determined using a BCA Protein Assay Kit (Thermo Fisher Scientific, Waltham, MA). Equal amounts of protein (10 μg per lane) were resolved on Mini-PROTEAN TGX precast gels (Bio-Rad, Hercules, CA) and transferred onto PVDF membranes. Membranes were blocked with 5% bovine serum albumin and incubated overnight at 4 °C with primary antibodies against phospho-STAT1, total STAT1, and β-actin (#9167S, #14994S, and #4970S, respectively; 1:1,000 dilution; Cell Signaling Technology, Danvers, MA). Following three washes with TBST buffer, membranes were incubated with HRP-conjugated goat anti-rabbit IgG (H+L) secondary antibody (#4050-05; 1:5,000 dilution; Southern Biotech, Birmingham, AL) for 1 h at room temperature. Protein bands were visualized using an Amersham Imager 680, and signal intensities were quantified using ImageJ software.

## Statistical analysis

Data are presented as mean±standard deviation (SD). The data related to qRT-PCR, disease scores, skin lesion scores, flow cytometry, Bio-Plex and chemistry parameters were analyzed with one-way ANOVA. Bacterial burdens between two groups were analyzed by the student t-test. Body weight changes were analyzed by two-way ANOVA. After a significant F-test for the ANOVA model, either Tukey's multiple comparisons test or Šídák's multiple comparisons test was used for comparison between groups. All data were analyzed by using GraphPad Prism software 10. Statistically significant values are denoted as * $p < 0.05$, ** $p < 0.01$, *** $p < 0.001$, and **** $p < 0.0001$, respectively. Comparisons with no significant differences are not labeled.

## Results

### hIFNG/hIFNGR mice are susceptible to *Ot* infection

Our previous study demonstrated that IFN-γ signaling is essential for host defense against *Ot* infection [24]. Given that human IFN-γ signaling may be less robust than its murine counterpart [33], we hypothesized that genetically engineered mice carrying human IFN-γ signaling may exhibit increased susceptibility to *Ot* infection. To test this, we utilized a genetically engineered humanized mouse model, hIFNG/hIFNGR, that possesses human IFN-γ signaling while lacking murine IFN-γ signaling. The strategy used to generate this mouse strain is shown in S1A Fig. The mRNA expression of human and mouse *Ifng*, *Ifngr1* and *Ifngr2* were confirmed by qRT-PCR (S1B-S1E Fig). The phosphorylation of STAT1 in T cells and B cells following IFN-γ stimulation ex vivo were analyzed by PhosFlow (S2A Fig). These data demonstrate the human IFN-γ signaling activation in hIFNG/hIFNGR mice. For the experiment of *Ot* infection, WT and *Ifngr1*[-/-] mice were also included as resistant and highly susceptible immunological control models, respectively [24]. We chose to use the intradermal infection route, as established in our previous studies [24,25], to closely mimic the natural mode of *Ot* transmission. As shown in Fig 1A and 1B, WT mice showed marginal body weight changes during infection with low disease scores. *Ifngr1*[-/-] mice were of high susceptibility, exhibiting a consistently decreasing body weight and increasing disease scores at 11 and 14 dpi. hIFNG/hIFNGR mice also showed similar body weight changes and disease scores as *Ifngr1*[-/-] mice until 11 dpi, but these mice started to gain body weight at 12 dpi and continued to recover. By 14 dpi, hIFNG/hIFNGR mice exhibited a comparable disease score as WT mice. Previously, we have shown the first report that *Ifngr1*[-/-] mice, but not WT mice, developed an eschar lesion at the skin inoculation site, which was characterized by necrosis and the formation of a black crust with a red halo, as seen in scrub typhus patients [24]. Here, we observed that hIFNG/hIFNGR mice also formed skin eschar lesions of comparable sizes as *Ifngr1*[-/-] mice at 11 dpi (Fig 1C and 1D). H&E staining of skin

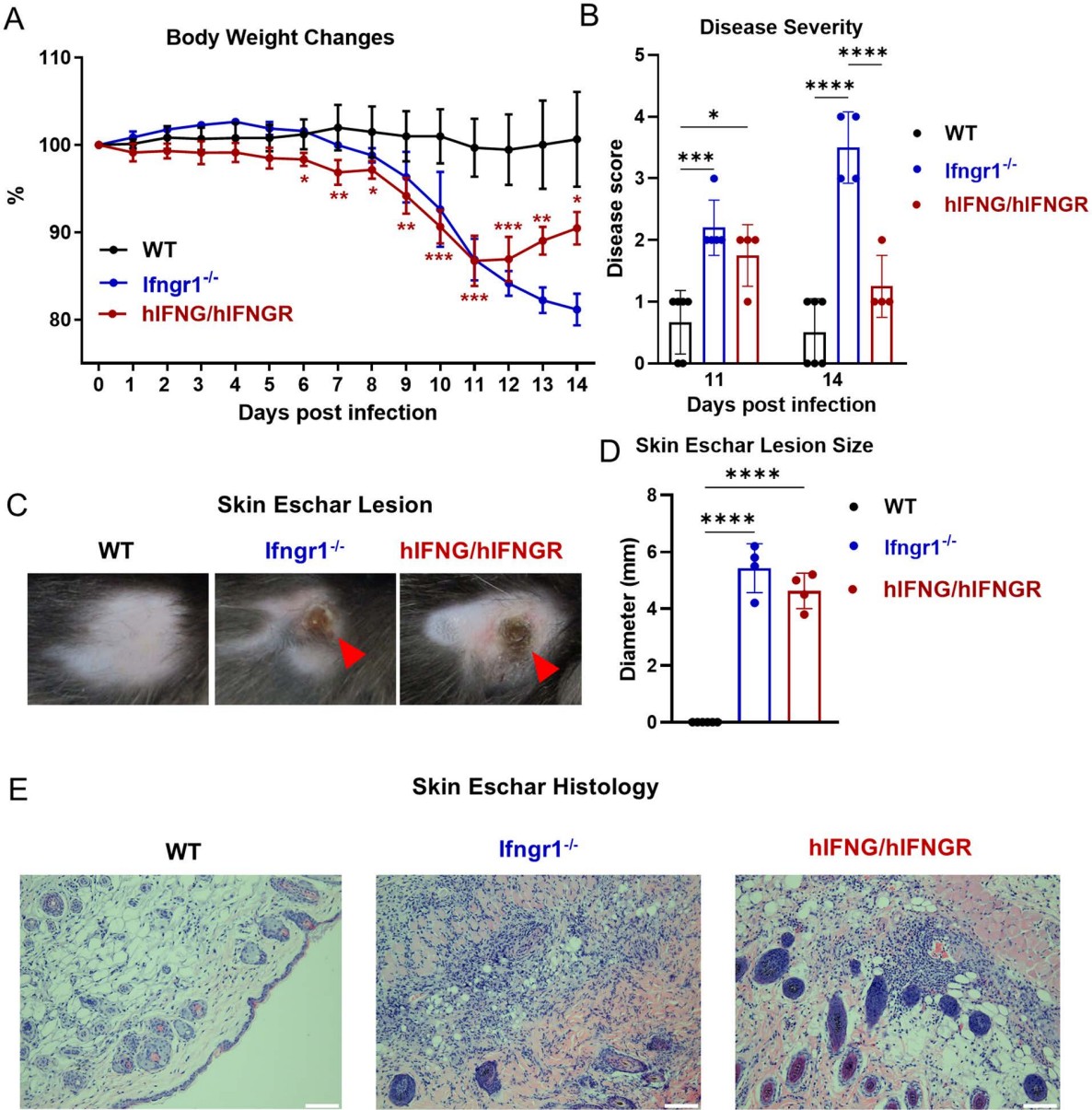

**Fig 1. hIFNG/hIFNGR mice are susceptible to *Ot* infection.** WT (n = 6), *Ifngr1⁻/⁻* (n = 4) and hIFNG/hIFNGR (n = 4) mice were intradermally infected with *Ot* (3 × 10³ FFU) on the right flank. Mice were monitored daily for (A) body weight changes and (B) disease scores. (C) Skin eschar lesions at inoculation sites at 11 dpi are presented and (D) their sizes are measured. (E) Representative histological images of the skin eschar lesion are shown at 11 dpi. Scale bar = 100 μm. Data are presented as mean ± SD from two independent pooled experiments. Body weight changes were analyzed by two-way ANOVA with Tukey's multiple comparisons between WT and hIFNG/hIFNGR mice at indicated time points. Disease scores (11 and 14 dpi) and skin eschar lesion sizes (11 dpi) were analyzed using one-way ANOVA followed by Tukey's multiple comparisons test to compare differences among mouse strains at each time point. *, $p < 0.05$; **, $p < 0.01$; ***, $p < 0.001$; ****, $p < 0.0001$. Comparisons with no significant differences are not labeled.

tissues in *Ifngr1⁻/⁻* and hIFNG/hIFNGR mice revealed extensive inflammatory cell infiltration, while WT mice without the presence of an eschar lesion exhibited minimal immune cell infiltration (Fig 1E). Therefore, our study provides the first evidence of skin eschar formation resembling human scrub typhus, and heightened susceptibility to *Ot* infection using a hIFNG/hIFNGR mouse model.

## Systemic inflammation in hIFNG/hIFNGR mice following *Ot* infection

To assess whether *Ot* infection induces inflammatory responses in major organs of hIFNG/hIFNGR mice, we performed histological analyses on the liver, lung, and brain tissues at 14 dpi (Fig 2). In uninfected mice, no notable immune cell infiltration was detected in the liver. Following *Ot* infection, WT mice showed mild inflammatory cell infiltration limited to the portal triads, with minimal involvement of the lobular regions. In contrast, hIFNG/hIFNGR mice exhibited moderate portal inflammation accompanied by inflammatory cell infiltration extending into the lobular areas. For the lung tissues, uninfected samples saw intact alveolar architecture, accompanied by minimal or no cellular infiltration. The lung histology of infected hIFNG/hIFNGR mice revealed increased mononuclear cell infiltration throughout the interstitium compared to infected WT mice, suggesting moderate interstitial pneumonitis in infected hIFNG/hIFNGR mice. It is known that *Ot* infection can cause blood brain barrier disruption and central nervous system disorders [34,39,40]. We also found that the meningeal layer of hIFNG/hIFNGR mice with *Ot* infection showed clear mononuclear cell infiltration, indicating meningitis or meningoencephalitis. Consistent with our previous report [24], *Ifngr1*$^{-/-}$ mice developed significant liver necrosis, extensive inflammatory infiltration in the lungs, and marked meningeal inflammation. Overall, our data demonstrated systemic inflammation across multiple organs in the hIFNG/hIFNGR mice following *Ot* infection.

## *Ot* infection induces human, but not murine IFN-γ in hIFNG/hIFNGR mice

To confirm the human IFN-γ signaling in hIFNG/hIFNGR mice following infection, we performed qRT-PCR to measure the transcript levels of mouse and human *Ifngr1* and *Ifngr2* in brain and lung tissues, which are the primary *Ot* target organs for severe disease outcomes [41]. We compared relative fold changes of receptor genes and found that hIFNG/hIFNGR mice express comparable or higher levels of receptor transcripts than WT mice, indicating intact human *Ifngr* expression before and after *Ot* infection (S2B Fig). We further quantified human IFN-γ protein levels in serum samples. We found that human IFN-γ levels in the serum were minimal in uninfected hIFNG/hIFNGR mice but increased significantly after infection, while human IFN-γ was undetectable in uninfected or infected WT mice (S2C Fig). Collectively, our result demonstrates that hIFNG/hIFNGR mice can produce human IFN-γ cytokine and receptors during *Ot* infection.

## Serum cytokines and biochemical profiles in hIFNG/hIFNGR mice with *Ot* infection

To better understand the status of systemic immune responses, we analyzed mouse serum cytokines by Bio-Plex assay (Fig 3A). Our result showed that mouse IFN-γ was significantly upregulated in infected WT mice, but it was undetected in hIFNG/hIFNGR mice as expected. Consistently high levels of IFN-γ were observed in infected *Ifngr1*$^{-/-}$ mice, as reported in our previous study [24], likely due to compensatory upregulation resulting from deficient IFN-γ signaling. A general trend of cytokine upregulation was observed in both WT and hIFNG/hIFNGR mice following infection, with increased levels of IL-2, IL-10 and IL-6. *Ifngr1*$^{-/-}$ mice produced limited IL-2, but substantial IL-6 and IL-10. Of these, serum IL-6 was significantly elevated in infected-hIFNG/hIFNGR mice when compared to WT controls, suggesting a heightened IL-6-mediated inflammatory response in the hIFNG/hIFNGR mouse model.

To evaluate animal physiological status, we measured serum biochemical parameters by using VetScan Comprehensive Diagnostic Profile reagent rotors. In the absence of infection, hIFNG/hIFNGR and *Ifngr1*$^{-/-}$ mice showed no physiological abnormalities, as indicated by serum biochemical parameters comparable to those of WT mice (Fig 3B). However, *Ot* infection resulted in significantly decreased levels of alkaline phosphatase (ALP), albumin (ALB), and glucose (GLU), with more pronounced reductions observed in hIFNG/hIFNGR mice. *Ifngr1*$^{-/-}$ mice also displayed lower levels of ALB and GLU. Amylase (AMY) levels showed no change in WT mice after infection, whereas a more pronounced decrease was observed in hIFNG/hIFNGR mice, consistent with findings in *Ifngr1*$^{-/-}$ mice [24]. These results may suggest compromised liver function and malnutrition in hIFNG/hIFNGR mice. In addition, a further increase of globulin (GLOB) in hIFNG/hIFNGR and *Ifngr1*$^{-/-}$ mice may indicate heightened inflammation in response to infection. Both WT and hIFNG/hIFNGR infected mice demonstrated a dysregulation of total protein (TP), alanine aminotransferase (ALT), phosphorus (PHOS)

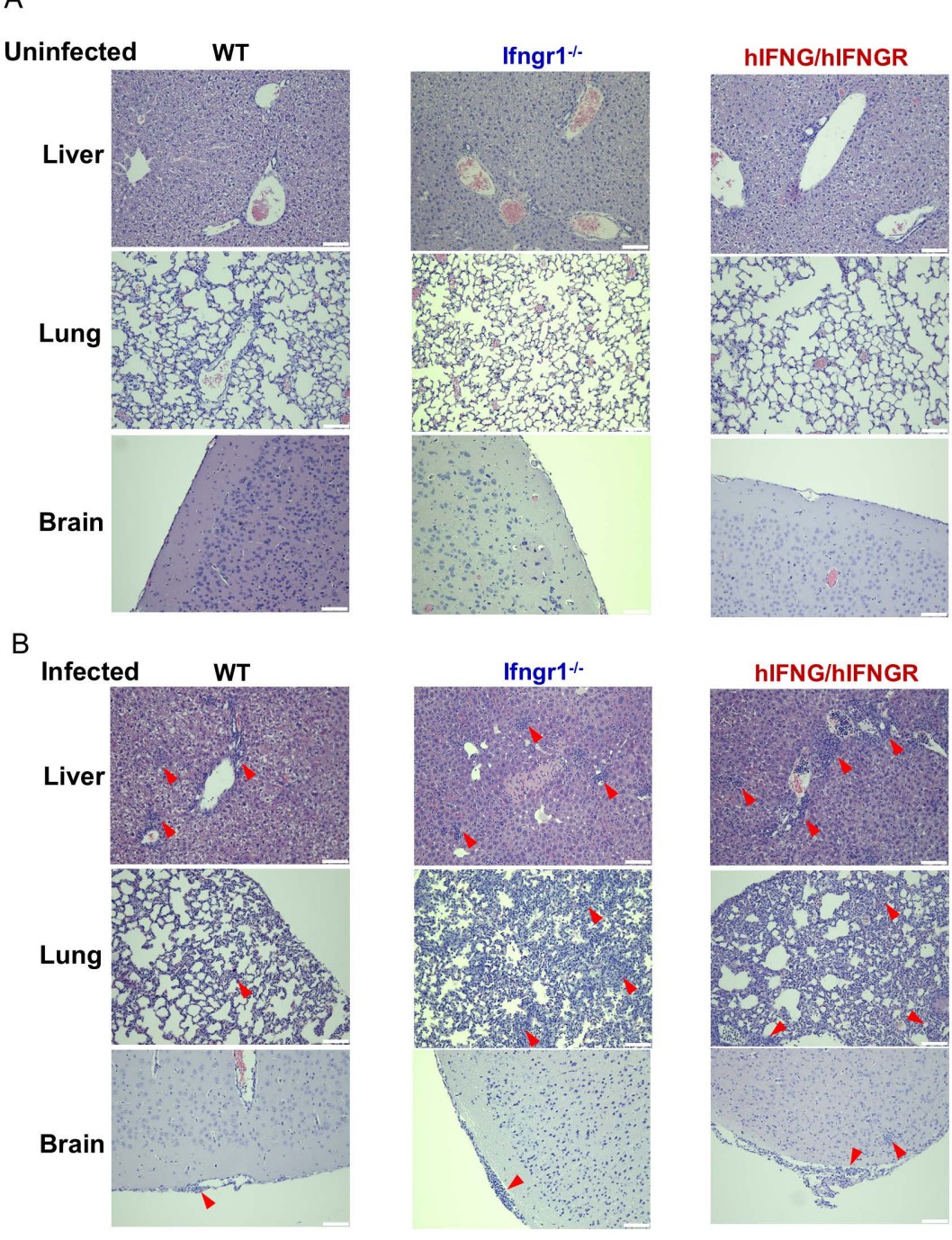

**Fig 2. Tissue histology of WT, *Ifngr1*⁻/⁻ and hIFNG/hIFNGR mice following *Ot* infection.** (A) Uninfected WT (n = 3), *Ifngr1*⁻/⁻ (n = 4) and hIFNG/hIFNGR (n = 2) mice were i.d. injected with L929 cell culture control. (B) WT (n = 9), *Ifngr1*⁻/⁻ (n = 4) and hIFNG/hIFNGR (n = 6) mice were infected as described in Fig 1. Representative histological images of the liver, lung and brain are shown at 14 dpi. The arrows indicate the focus of inflammatory infiltration in the tissues. Scale bar = 100 μm.

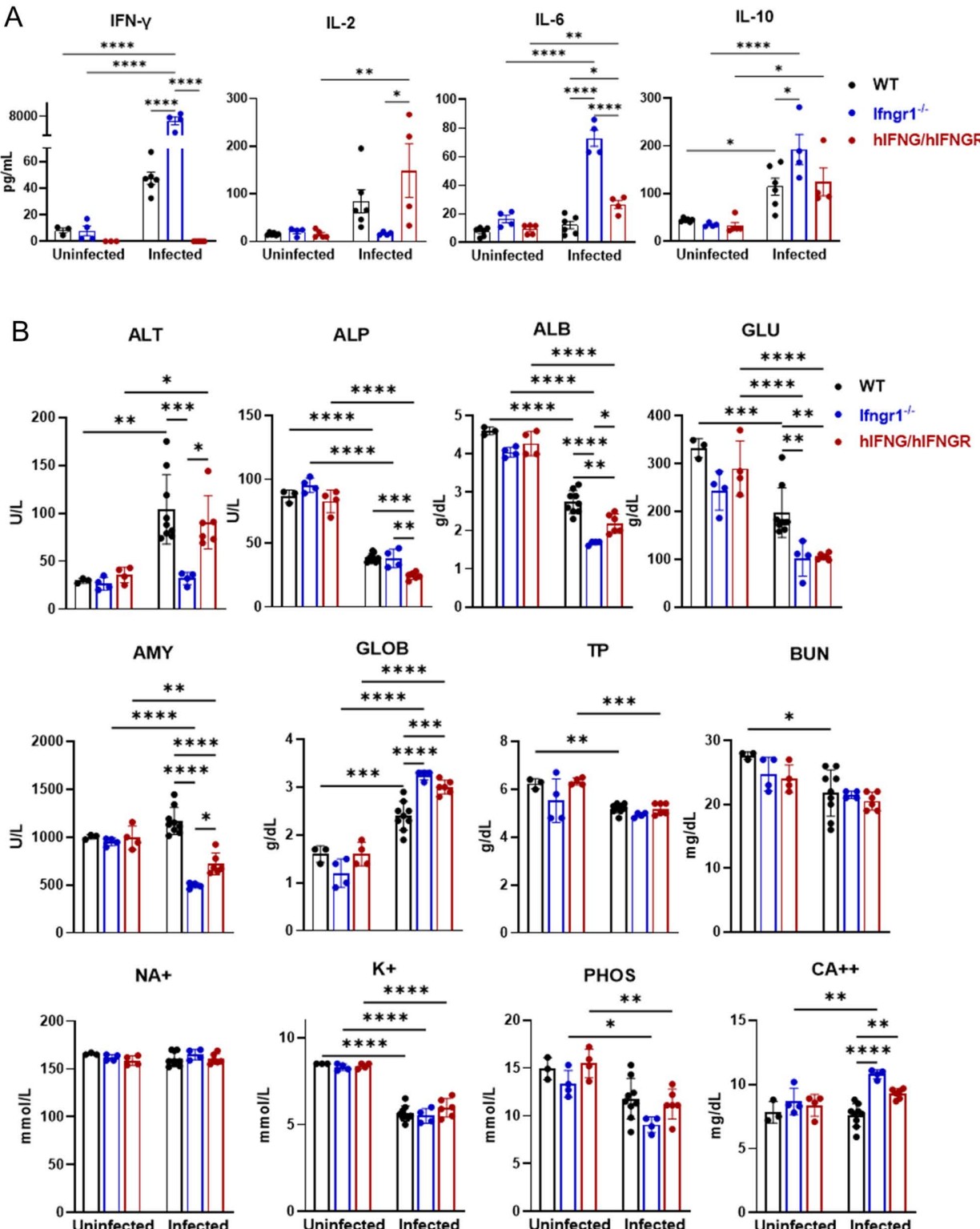

**Fig 3. Serum cytokines and chemistry profiles in mice with *Ot* infection.** Mice were infected as described in Fig 1. (A) Serum cytokine levels were analyzed at 14 dpi by a Bio-Plex assay. (B) Serum chemistry parameters were detected by using VetScan Comprehensive Diagnostic Profile Reagent Rotor. The parameters include alanine aminotransferase (ALT), albumin (ALB), alkaline phosphatase (ALP), amylase (AMY), total calcium (CA++),

globulin (GLOB), glucose (GLU), potassium (K⁺), sodium (NA⁺), total protein (TP), blood urea nitrogen (BUN), and phosphorus (PHOS). Data are shown as mean±SD from three pooled independent experiments and analyzed by one-way ANOVA with a Tukey's multiple comparisons test. Comparisons were displayed among mouse strains under either uninfected or infected conditions. In addition, comparisons were displayed within each mouse strain between uninfected and infected conditions. *, $p < 0.05$; **, $p < 0.01$; ***, $p < 0.001$; ****, $p < 0.0001$. Comparisons with no significant differences are not labeled.

and potassium (K+). These findings suggest a systemic inflammatory response and disrupted physiological homeostasis in hIFNG/hIFNGR mice following *Ot* infection.

To assess whether hIFNG/hIFNGR mice exhibit altered humoral immunity in response to *Ot* infection, we measured *Ot*-specific antibody titers in mouse sera. ELISA results showed comparable levels of IgM and IgG against *Ot* TSA56 antigen between hIFNG/hIFNGR and WT mice (S3 Fig), indicating that hIFNG/hIFNGR mice mount a competent humoral immune response like WT controls. Notably, *Ifngr1⁻/⁻* mice produced lower levels of IgG following *Ot* infection compared with WT mice, suggesting that this IFN-γ signaling-deficient strain may be suboptimal for evaluating humoral immune responses and vaccine candidates.

### Elevated bacterial burden but reduced interferon-stimulated gene (ISG) expression in hIFNG/hIFNGR mice

To assess bacterial control in hIFNG/hIFNGR mice, bacterial burdens were measured in various organs (Fig 4A). Consistent with previous findings [21,24], the highest bacterial burden was observed in mouse lungs, followed by the spleens in both WT and hIFNG/hIFNGR mice. We found that hIFNG/hIFNGR mice exhibited approximately a 2-fold increase in bacterial burden in the lungs and spleen. Notably, a 10-fold increase of bacterial burdens in the brain was detected in hIFNG/hIFNGR mice as compared to WT mice. Likewise, the kidney and liver organs in hIFNG/hIFNGR mice also showed significantly higher bacterial burden. In addition, *Ifngr1⁻/⁻* mice failed to control bacterial replication in various organs (Fig 4B) as we observed previously [24]. Therefore, these results suggest that hIFNG/hIFNGR mice confers attenuated antibacterial activity compared to WT controls.

IFN-γ signaling activates the JAK-STAT pathway and induces the expression of multiple downstream ISGs, driving a robust antibacterial response [42]. Disruption of IFN-γ/STAT1 signaling has been shown to result in uncontrolled bacterial replication and lethal *Ot* infection [24]. To confirm the human IFN-γ signaling during *Ot* infection, we assessed phosphorylated STAT1 in lung tissues following *Ot* infection. We found a reduced expression of phosphorylated STAT1 in the lung of hIFNG/hIFNGR mice as compared to WT mice (S4 Fig), indicating that hIFNG/hIFNGR mice can produce human IFN-γ and activate downstream signaling following *Ot* infection, although this activation is attenuated compared with WT mice. We further examined ISG transcript levels in various organs by qRT-PCR. Our results demonstrated that as compared to WT mice, hIFNG/hIFNGR mice showed lower expression of all examined ISGs in the brain, including *Cxcl10*, *Nos2*, *Ifit1/2*, *Gbp2/4/5*, *Isg15*, *Isg20*, *Stat1*, *Rsad2*, *Oas3*, *Oas1b* and *Oasl2* (Fig 4C). Furthermore, we confirmed that the lungs of hIFNG/hIFNGR mice also exhibited lower expression of some ISGs including *Cxcl10*, *Ifit1/2*, *Rsad2*, *Isg15* and *Isg20* (S5 Fig). As expected, *Ifngr1⁻/⁻* mice exhibited significantly reduced ISG expression, particularly for *Stat1* and *GBP* family genes (Figs 4C and S5 Fig). In addition, the inflammatory cytokine levels (*IL6*, *Tnf* and *Il27)* were comparable in both lungs and brain between WT and hIFNG/hIFNGR (Figs 4C and S5 Fig). Interestingly, we found that hIFNG/hIFNGR mice expressed significantly higher levels of *Sele* (E-selectin) and *Selp* (P-selectin) in the brain, but not in the lungs, indicating increased endothelial activation and inflammation in the brain of hIFNG/hIFNGR mice with *Ot* infection.

### Dysregulated innate and adaptive immune cell response in hIFNG/hIFNGR mice following *Ot* infection

*Ot* infection triggers robust innate and adaptive immune responses, characterized by strong type 1 immunity and activation of multiple immune cell subsets, including NK cells, macrophages, dendritic cells, neutrophils, and effector T cells.

PLOS Pathogens

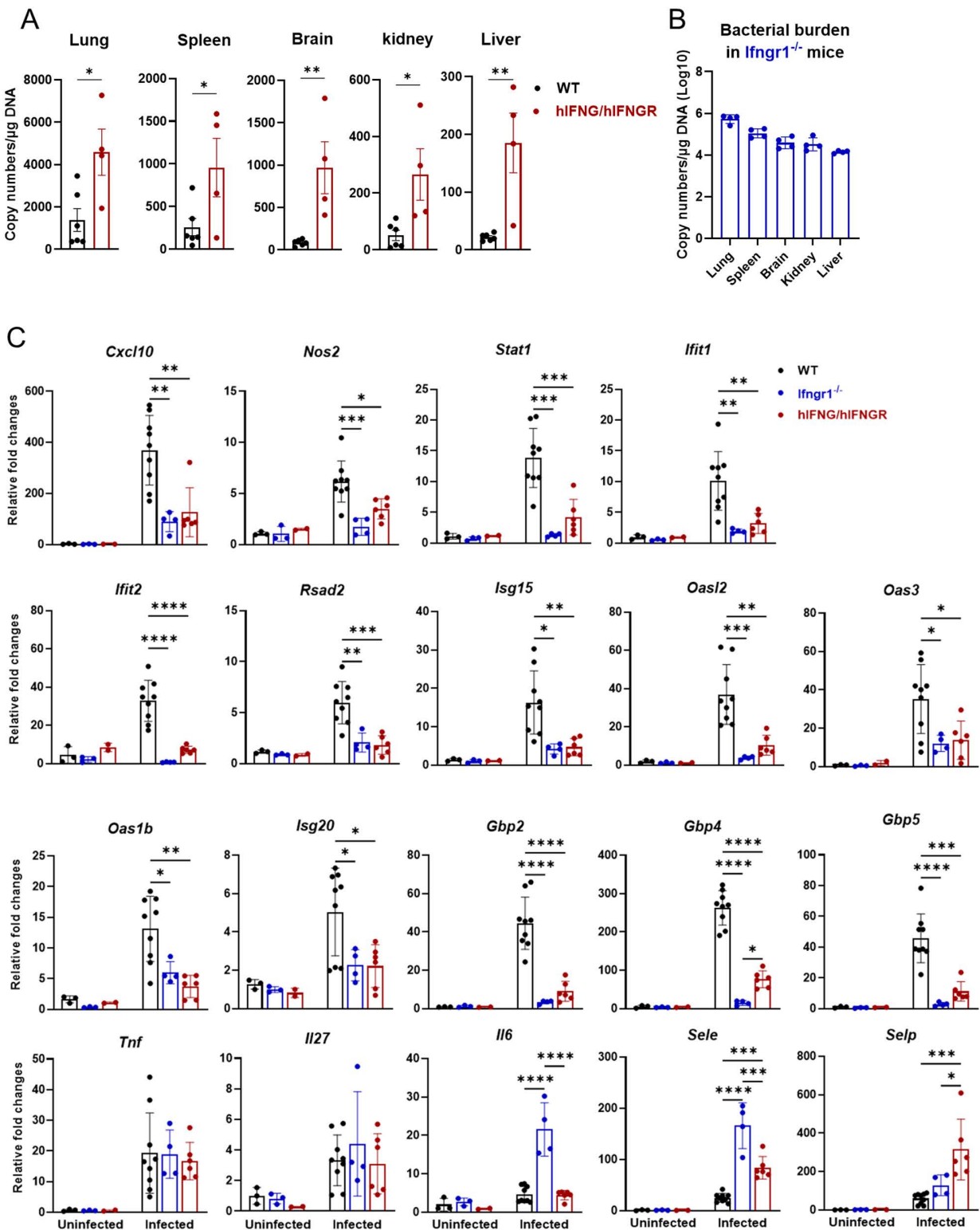

**Fig 4. hIFNG/hIFNGR mice exhibit an increased bacterial burden and reduced expression of interferon stimulated genes (ISGs) following *Ot* infection.** Mice were infected as described in Fig 1 and were euthanized at 14 dpi. Genomic DNA was isolated from the lung, spleen, brain, kidney and liver of (A) infected WT, hIFNG/hIFNGR and (B) *Ifngr1*-/- mice. Bacterial burden was measured by qPCR. (C) The transcript levels of ISGs in the brain

were analyzed by qRT-PCR. Data is shown as mean ± SD from three pooled independent experiments. Unpaired t-test was performed for bacterial burdens. ISG expression data of three infected groups were analyzed by one-way ANOVA with a Šídák's multiple comparisons test. *, $p < 0.05$; **, $p < 0.01$; ***, $p < 0.001$; ****, $p < 0.0001$. Comparisons with no significant differences are not labeled.

While these cells play critical roles in controlling bacterial replication and dissemination, their excessive activation can lead to immunopathogenesis. We found that *Ot* infection induced stronger CD4+ and CD8+ T effector cell responses in the spleens of hIFNG/hIFNGR mice compared to WT controls (Fig 5A and 5B). However, there was a significantly decreased percentage of Treg cells among CD4+ T cells (S6 Fig). Moreover, CTLA4+Treg cell numbers were markedly reduced in hIFNG/hIFNGR mouse spleens (S6 Fig). This result suggests that *Ot* infection leads to excessive T cell activation accompanied by a diminished immunosuppressive T cell response in hIFNG/hIFNGR mice. NK cells not only produce IFN-γ during *Ot* infection but are also activated by IFN-γ signaling [24,43–45]. We found that as compared to WT controls, hIFNG/hIFNGR mice harbored significantly fewer NK cells, which displayed a markedly lower activation status, as evidenced by reduced CD69 expression (Fig 5C and 5D). Similarly, the impaired NK cell activation was also observed in *Ifngr1*-/- mice as expected.

Neutrophilia is observed in scrub typhus patients and is linked to disease progress [46,47]. We found that infected hIFNG/hIFNGR and *Ifngr1*-/- mice exhibited much higher neutrophil numbers in the spleen as compared to WT controls (Fig 6A). Ly6chi monocytes were also recruited into the spleen following infection, with higher infiltration levels observed in hIFNG/hIFNGR mice (Fig 6B). Notably, both macrophages and dendritic cells in infected hIFNG/hIFNGR and *Ifngr1*-/- mice exhibited reduced MHC II expression (Fig 6C), indicating impaired activation of antigen-presenting cells in response to attenuated or impaired IFN-γ signaling, respectively. Overall, our results suggest an increased neutrophil response alongside impaired activation of other innate immune cells in hIFNG/hIFNGR mice following *Ot* infection.

## Discussion

Scrub typhus is a systemic, life-threatening disease with a high incidence in the "tsutsugamushi triangle", yet it remains significantly neglected [48]. A better understanding of *Ot* pathogenesis and the development of effective vaccines represent the most promising strategies for preventing future outbreaks of this disease [49,50]. Although several animal models have been established by our group and other researchers [13–18,20,21,27,51,52], there remains a need for an accurate mouse model that closely recapitulates human scrub typhus. In this study, we introduce a newly developed genetically engineered humanized mouse model for scrub typhus, featuring human IFN-γ signaling in the absence of murine IFN-γ signaling. We demonstrated that this novel mouse strain exhibits increased susceptibility to intradermal *Ot* infection, characterized by multiorgan bacterial dissemination, skin eschar formation, systemic inflammation, altered serum biochemical profiles, and dysregulated innate and adaptive immune responses. This study represents the first establishment of a scrub typhus model using genetically engineered human IFN-γ signaling mouse strain, and our findings support the utility of this model as a promising tool for future studies on pathogenesis, drug screening, and vaccine development.

We have previously demonstrated that IFN-γ signaling is necessary for host defense against *Ot* infection as the *Ifngr1*-/- mouse strain, in contrast to WT mice, develops a lethal model of scrub typhus and forms a skin eschar lesion at the site of intradermal inoculation [24]. The eschar serves as a key diagnostic indicator in patients with acute febrile illness in regions endemic for scrub typhus [53–55]. Clinical observation indicates that the presence of an eschar might be associated with increased disease severity [56,57]. Our study reports skin eschar lesions in hIFNG/hIFNGR mice in response to *Ot* infection for the first time. Representative HE staining of eschar lesions showed apparent immune infiltration in both hIFNG/hIFNGR and *Ifngr1*-/- mice, but not in WT mice (Fig 1E). Therefore, our data generated from hIFNG/hIFNGR mouse model, along with the previously published *Ifngr1*-/- model [24] demonstrate that IFN-γ signaling is a critical determinant of eschar formation and disease severity of scrub typhus. The downstream mechanisms by which IFN-γ signaling influences eschar formation, however, remain unclear. Interestingly, unlike mouse cells that express strong inducible nitric oxide synthase

 

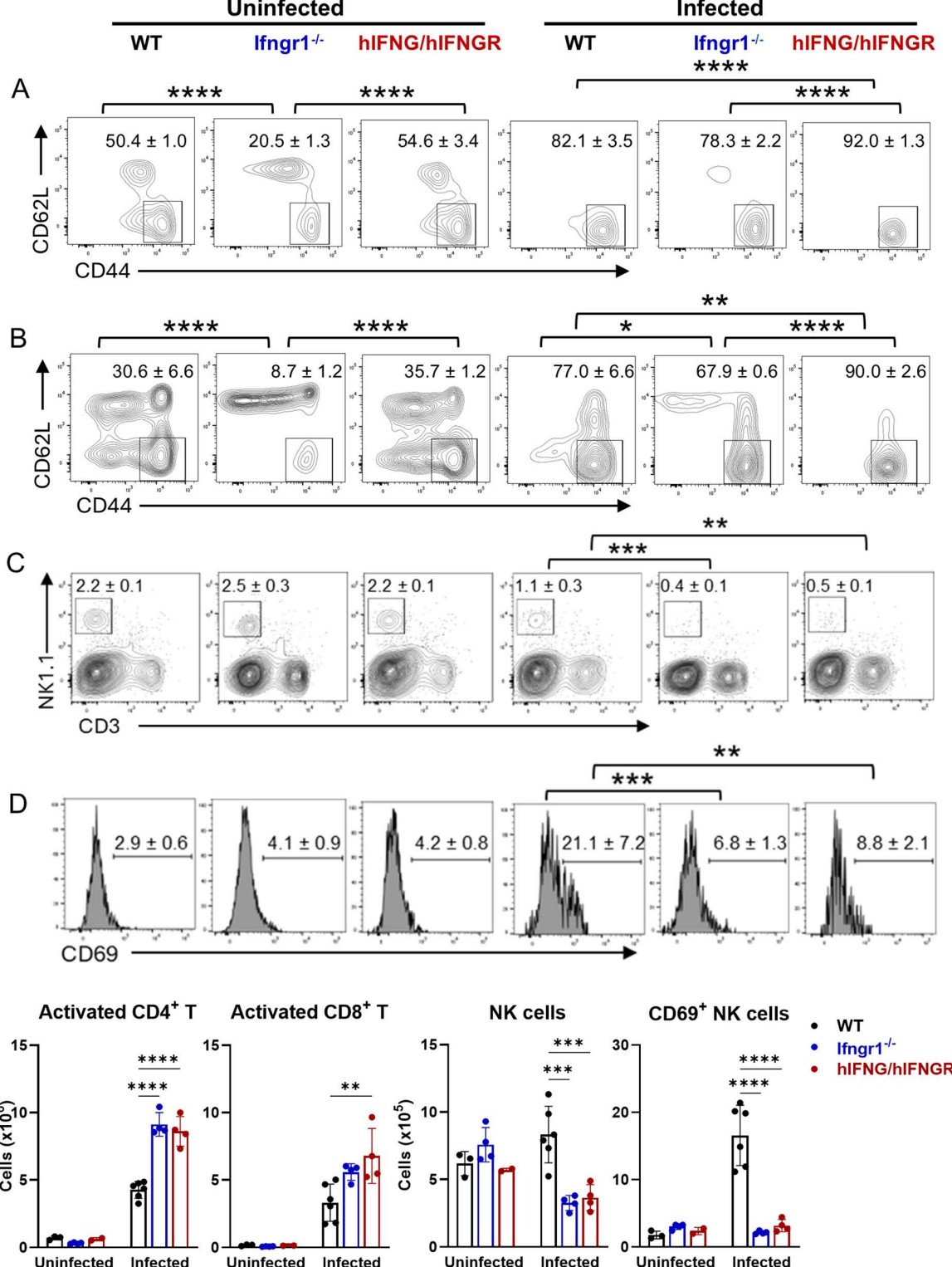

**Fig 5. Imbalanced T cell and NK cell responses in hIFNG/hIFNGR mice following infection.** Mice were infected as described in Fig 1 and were euthanized at 14 dpi. Spleen cells were isolated and analyzed by flow cytometry. Single cells were gated first by FSC/SSC, followed by the exclusion of dead cells by live/dead staining. CD4+ and CD8+ T cells were gated on CD3+CD4+ and CD3+CD8+, respectively. CD44+CD62L- T cells were identified as

activated populations on (A) CD4$^+$ and (B) CD8$^+$ T cells. (C-D) NK cells were gated on CD3$^-$NK1.1$^+$, and CD69 was used as an activation marker of NK cells. The percentages of cell populations were shown as mean±SD on the flow cytometric images and the statistical analysis between groups were labeled. The absolute numbers of cell populations were also calculated and shown below the flow cytometric images. One-way ANOVA with a Šídák's multiple comparisons test was used for data analysis. Comparisons were displayed among mouse strains under either uninfected or infected conditions. *, $p < 0.05$; **, $p < 0.01$; ***, $p < 0.001$; ****, $p < 0.0001$. Comparisons with no significant differences are not labeled.

(iNOS) by IFN-γ stimulation, human cells instead produce indoleamine dioxygenase [33]. Due to the key role of iNOS in wound healing [58], it is possible that compromised IFN-γ/iNOS signaling contributes to eschar formation in not only scrub typhus, but also rickettsial infection [59]. Further studies are needed to elucidate downstream mechanisms by which IFN-γ signaling influences eschar formation.

Scrub typhus can cause systemic infection and multiorgan dysfunction, particularly in cases with delayed diagnosis and treatment. Immune infiltration in response to infection leads to inflammatory damage and organ failure in patients [54,60–62]. Our histology assessment, serum Bio-plex assay, biochemical analysis and bacterial burden measurement collectively demonstrate systemic bacterial dissemination and inflammatory responses across multiple organs, including lung, liver and brain in the hIFNG/hIFNGR mouse model (Figs 2, 4 and 5). The immune and biochemical profiles revealed unique features in hIFNG/hIFNGR mice that were consistent with those observed in *Ifngr1$^{-/-}$* group [24]. For example, serum analysis revealed elevated IL-6 in hIFNG/hIFNGR and *Ifngr1$^{-/-}$* mouse model mice compared to WT mice following infection, a characteristic that is also noted in scrub typhus patients [63,64], indicating that the proinflammatory cytokine IL-6 may serve as a potential biomarker for predicting disease severity in scrub typhus. In addition, the decreased serum levels of ALB, GLU and AMY which were found in infected *Ifngr1$^{-/-}$* mice, were also observed in hIFNG/hIFNGR mice compared to WT controls (Fig 3B), indicating excessive liver dysfunction in this mouse model. Therefore, combining serum cytokine profiles and biochemical parameters with bacterial load may offer a more accurate prediction of disease severity in scrub typhus patients.

Scrub typhus has become a leading cause of central nervous system infection in endemic regions [65,66]. Severe infection can lead to a range of neurological complications, including meningitis, encephalitis, and meningoencephalitis, which are associated with increased mortality [34,40,67–70]. Moreover, *Ot* infection may increase the risk of long-term sequelae and neurodegenerative diseases, such as dementia [71,72]. In this study, the increased susceptibility of hIFNG/hIFNGR mice to *Ot* infection is evidenced by higher bacterial burdens in multiple organs compared to WT mice (Fig 4A). Notably, brain tissues of hIFNG/hIFNGR mice exhibited the most pronounced increase, with bacterial burdens approximately 10-fold higher than those in WT controls. The uncontrolled bacterial replication observed in the brain may be attributed to the substantial downregulation of several ISGs (Fig 4B). Consistent with this, histological analysis also showed significant immune cell infiltration within the brain meninges of hIFNG/hIFNGR mice, which may contribute to meningoencephalitis (Fig 2). The significantly elevated expression of key adhesion molecules *Sele* and *Selp* in the brains of hIFNG/hIFNGR mice may indicate robust neuroinflammation and potential blood-brain barrier dysfunction [73–75]. These findings may support the utility of the hIFNG/hIFNGR mouse model as a valuable tool for investigating neurological complications associated with scrub typhus.

We further evaluated innate and adaptive immune responses following infection. Activation of IFN signaling induces robust expression of ISGs, which play a critical role in controlling pathogen replication [76–78]. Thus, we first analyzed a comprehensive ISG expression panel in both the brain and lungs. We found that the expression of several ISGs (*Ifit1, Ifit2, Rsad2, Isg15, Cxcl10*) were downregulated in both organs of hIFNG/hIFNGR mice, whereas members of the GBP family (*Gbp2, Gbp4, Gbp5*), were selectively reduced in the brain. Given that bacterial burdens in hIFNG/hIFNGR mice were approximately 10-fold higher in the brain and only 2.5-fold higher in the lungs compared to WT mice (Fig 4A), these findings suggest that distinct ISGs may contribute differentially to bacterial control in an organ specific manner. *Ifngr1$^{-/-}$* mice exhibited the downregulation of most ISGs, particularly the GBP family. Next, we used flow cytometry to profile innate and

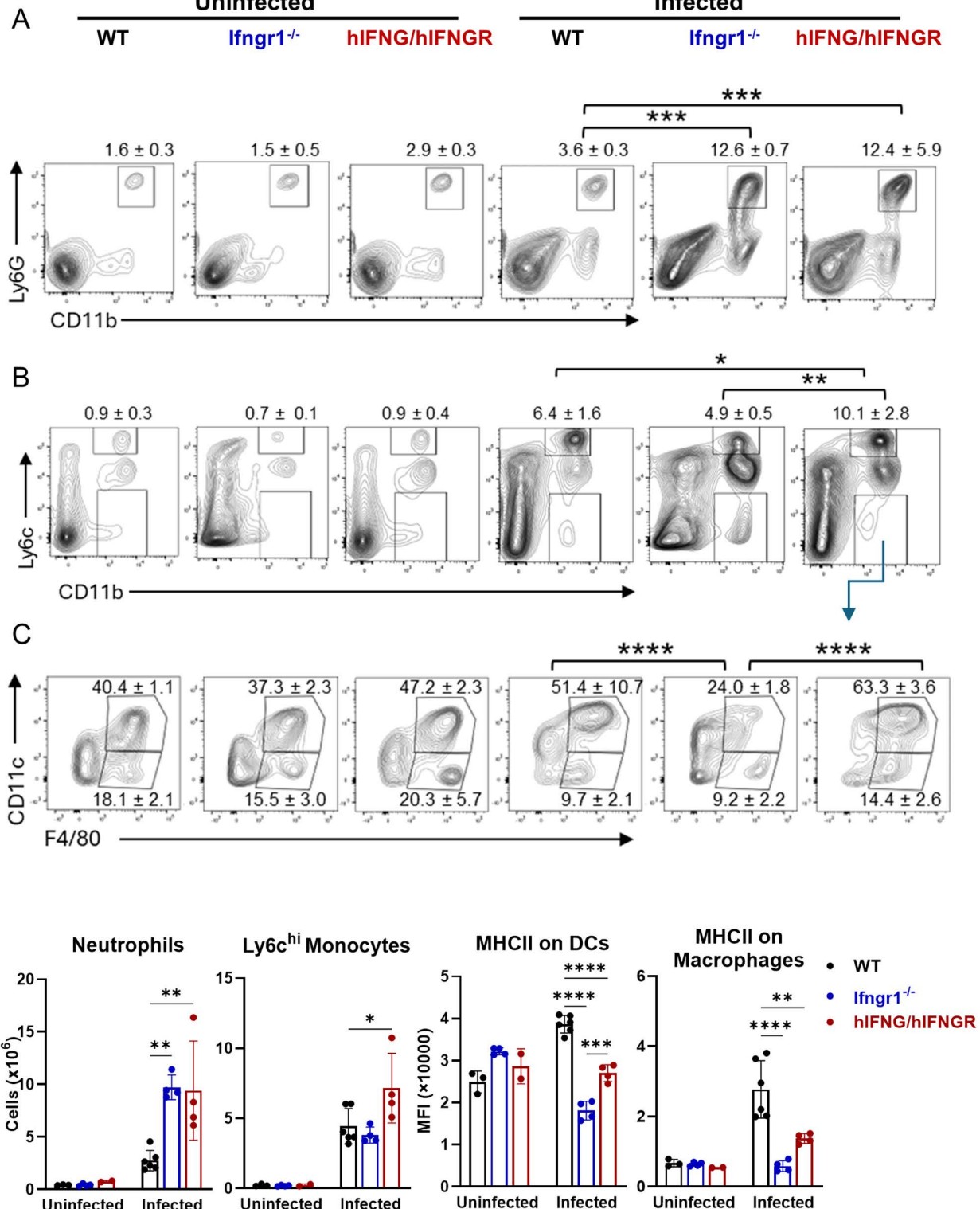

**Fig 6. Excessive neutrophils and reduced innate immune cell activation in hIFNG/hIFNGR mice following *Ot* infection.** Mice were infected as described in Fig 1 and were euthanized at 14 dpi. Spleen cells were isolated and analyzed by flow cytometry. Single cells were gated first by FSC/SSC, followed by the exclusion of dead cells by live/dead staining. (A-B) Neutrophils and monocytes were identified by CD11b⁺Ly6G⁺ and CD11b⁺Ly6Cʰⁱ,

respectively. (C) CD11b+Ly6C- cells were further gated by CD11c and F4/80 markers. Macrophages and dendritic cells were characterized by CD11c-F4/80+ and CD11chiF4/80+, respectively. The mean fluorescent intensity (MFI) of MHCII on macrophages and dendritic cells were measured. The percentages of cell populations were shown as mean±SD on the flow cytometric images and the statistical analysis between infected groups were labeled. The absolute numbers and MFI of cell populations were also calculated and were shown below the flow cytometric images. One-way ANOVA with a Šídák's multiple comparisons was used for data analysis. Comparisons were displayed among mouse strains under either uninfected or infected conditions. *, $p < 0.05$; **, $p < 0.01$; ***, $p < 0.001$; ****, $p < 0.0001$. Comparisons with no significant differences are not labeled.

adaptive immune cell populations in the spleens. We observed an imbalance between effector and regulatory T cells in hIFNG/hIFNGR mice, characterized by an increase in activated effector T cells and a reduction in Treg cells (Figs 5A-5B and S6). This imbalance might be due to the impaired bacterial clearance and may lead to heightened systemic inflammation. It is known that IFN signaling is critical for antigen-presenting cell maturation and NK cell activation [45,79–81]. Our data showed that humanized IFN-γ signaling led to a significant reduction in MHC class II expression on antigen-presenting cells and impaired activation of NK cells (Figs 5-6). Neutrophilia is a common feature of acute scrub typhus [46] and increased neutrophil activation has been associated with disease progression [47]. Consistent with these clinical observations, we found that neutrophil populations were more than 3-fold higher in hIFNG/hIFNGR mice compared to WT mice following infection (Fig 6), suggesting that this hIFNG/hIFNGR mouse model recapitulates key aspects of immune dysregulation observed in patients with acute scrub typhus. Lastly, we measured the *Ot*-specific antibody titers in mouse serum and found that hIFNG/hIFNGR, but not the *Ifngr1-/-* mice produced IgM and IgG comparable to those of WT controls (S3 Fig). The intact antibody response supports the suitability of this animal model for future vaccine studies.

Humanized mouse models represent promising tools for studying scrub typhus. Jiang et al. found that humanized DRAGA mice were highly susceptible to intradermal and subcutaneous infection, mounted strong CD4+ and cytotoxic T cell responses, and produced human IgM and IgG following repeated immunization, suggesting it as a new pre-clinical model for *Ot* pathogenesis study and vaccine candidate testing [52]. Our study using mice with human IFN-γ signaling builds on our recent findings that IFN-γ, rather than type I interferons, plays a critical role in skin eschar formation and host defense against *Ot* infection [24]. To our knowledge, this triple KO/KI mouse model has never been applied in any research study so far. To confirm that human IFN-γ signaling is functional in this mouse strain, we present multiple lines of evidence generated by Biocytogen or by our laboratory (S2 and S4 Figs). First, qRT-PCR data demonstrates that hIFNG/hIFNGR mice only express human *Ifngr1* and *Ifngr2* transcripts. The levels of these transcripts are comparable or higher than those of mouse counterparts in WT mice. Secondly, PhosFlow data show that immune cells (e.g., T and B cells) from hIFNG/hIFNGR mice can respond to recombinant human, but not mouse IFN-γ as evidenced by increased STAT1 phosphorylation. Thirdly, our ELISA data demonstrates significantly increased human IFN-γ levels in hIFNG/hIFNGR mice following infection. Lastly, our western blot result confirms the expression of phosphorylated STAT1 in hIFNG/hIFNGR after infection. Notably, we also found that the phosphorylated STAT1 signaling is attenuated in hIFNG/hIFNGR mice as compared to WT mice. It is reported that human IFNGR expressed in mouse macrophages can bind human IFN-γ with an affinity comparable to that observed in human cells, and ligand binding elicits canonical IFN-γ-responsive signaling [82]. However, human IFN-γ fails to fully protect mouse cells from the cytopathic effects of encephalomyocarditis virus and vesicular stomatitis virus, whereas mouse IFN-γ provides robust protection [82]. The mechanisms underlying this species-specific IFN-γ response remain unclear and may involve differences in receptor binding affinity, JAK1/JAK2 association, or STAT1 recruitment and phosphorylation. Further studies are warranted to reveal the underlying mechanism of species-specific IFN-γ regulation. Collectively, our findings demonstrate that hIFNG/hIFNGR mice retain functional but attenuated IFN-γ signaling, which may underlie their relative susceptibility to *Ot* infection.

A limitation of our study is the limited number of animals available, which restricted our ability to comprehensively assess the temporal dynamics of susceptibility, pathology, and immune responses. Since our model is a genetically engineered humanized mouse strain, it should not be confused with cellular humanization, where mice are engrafted with human cells such as hematopoietic stem cells or peripheral blood mononuclear cells. Moreover, these germline-modified mice can be

bred, making them a more cost-effective and time-efficient option for experimental studies. In addition, this mouse strain may serve as a useful model for evaluating vaccines and therapeutics targeting pathogens that rely heavily on IFN-γ-mediated immunity such as rickettsial infection [83–85] as well as other bacteria, protozoa, fungi and viruses [86].

In sum, we reported a novel mouse model that is susceptible to *Ot* infection and forms skin eschars via an intradermal infection route, effectively replicating human scrub typhus disease. Future investigations are warranted by using this novel model to evaluate bacterial virulence, characterize host protective immunity and assess the efficacy of potential new vaccine candidates.

## Supporting information

**S1 Table. Sequences of PCR primers.** All primer sequences are obtained from PrimerBank (https://pga.mgh.harvard.edu/primerbank/) or our previous publications.
(DOCX)

**S1 Fig. Strategy of generating hIFNG/hIFNGR mice. hIFNG/hIFNGR mice were generated by Biocytogen.** (A) Schematic strategy for generation of humanized IFNG, IFNGR1, and IFNGR2 alleles. (B) *Ifng* and *Ifngr1* expression was analyzed in B-hIFNGR1/hIFNG mice by RT-PCR. Human *Ifng* and *Ifngr1* mRNA were detectable in splenocytes of the homozygous B-hIFNGR1/hIFNG mice (H/H), but not in WT mice (+/+). (C) Human *Ifngr2* mRNA was exclusively detectable in the small intestine of homozygous B-hIFNGR2mice (H/H), but not in that of WT mice (+/+). (D-E) PCR primers and experimental conditions were listed.
(TIF)

**S2 Fig. hIFNG/hIFNGR mice display functional, but attenuated IFN-γ signaling.** (A) Splenocytes were isolated from naïve hIFNG/hIFNGR mice, followed by mouse or human IFN-γ cytokine stimulation ex vivo. The phosphorylated STAT1 protein expression in T and B cells was analyzed by PhosFlow. (B) Mice were infected as in Fig 1. Brain and lung tissues were harvested for analyzing mouse and human *Ifngr1* and *Ifngr2* transcripts by qRT-PCR. Unpaired t-test was used for data analysis between two groups under either uninfected or infected condition. *, $p < 0.05$; **, $p < 0.01$; ***, $p < 0.001$; ****, $p < 0.0001$. (C) Serum samples were collected from WT and hIFNG/hIFNGR mice at 14 dpi. Human IFN-γ levels were measured by ELISA. One-way ANOVA with a Šídák's multiple comparisons was used for data analysis. ****, $p < 0.0001$, u.d., undetectable. Data are presented as mean ± SD from three independent pooled experiments. Comparisons with no significant differences are not labeled.
(TIF)

**S3 Fig. Comparable *Ot*-specific IgM and IgG levels between hIFNG/hIFNGR and WT control mice.** To measure relative amounts of IgM and IgG antibodies, mouse sera were collected at 14 dpi as shown in Fig 1 and then diluted to create a dilution curve. Uninfected mouse sera were used as controls. The area under the curve (AUC) is calculated from three pooled independent experiments and shown as mean ± SD. One-way ANOVA with a Šídák's multiple comparisons was used for data analysis of infected groups. *, $p < 0.05$.
(TIF)

**S4 Fig. Phosphorylated STAT1 expression in the lung following *Ot* infection.** Mice were infected as described in Fig 1 and were euthanized at 14 dpi. Lung tissues were collected and analyzed for phospho-STAT1, total STAT1, and β-actin by western blot. Protein bands were visualized using an Amersham Imager 680, and signal intensities were quantified using ImageJ software. Data is shown as mean ± SD and analyzed by one-way ANOVA with a Šídák's multiple comparisons test. *, $p < 0.05$; **, $p < 0.01$; ***, $p < 0.001$, ****, $p < 0.0001$. Comparisons with no significant differences are not labeled.
(TIF)

**S5 Fig. Reduced expression of ISGs in the lungs of hIFNG/hIFNGR mice with *Ot* infection.** Mice were infected as described in Fig 1 and were euthanized at 14 dpi. The transcript levels of ISGs and inflammatory genes in the lungs were analyzed by qRT-PCR. Data is shown as mean ± SD from three pooled independent experiments. One-way ANOVA with a Šídák's multiple comparisons test was performed for the infected groups. *, $p < 0.05$; **, $p < 0.01$; ***, $p < 0.001$. Comparisons with no significant differences are not labeled.
(TIF)

**S6 Fig. Reduced Treg cells in hIFNG/hIFNGR mice following *Ot* infection.** Mice were infected as described in Fig 1 and were euthanized at 14 dpi. Splenocytes were isolated for flow cytometric analysis of regulatory T (Treg) cells, which were identified by CD4+ Foxp3+. The expression of CTLA4 was further gated on CD4+Foxp3+ T cells. The percentages of cell populations were shown as mean ± SD on the flow cytometric images and the statistical analysis between infected groups were labeled. The absolute numbers of cell populations were calculated and were shown below the flow cytometric images. One-way ANOVA with a Šídák's multiple comparisons was used for data analysis of infected groups. *, $p < 0.05$; **, $p < 0.01$; ***, $p < 0.001$; ****, $p < 0.0001$. Comparisons with no significant differences are not labeled.
(TIF)

## Acknowledgments

We would like to thank the UTMB Flow Cytometry and Cell Sorting Core Lab (Meredith Weglarz) and Dr. Joseph Jelinski for sample analysis and manuscript revision, respectively.

## Author contributions

**Conceptualization:** Yuejin Liang.

**Data curation:** Ryan H Cho, Lihai Gao, Hui Wang, Yixuan Zhou, Casey Gonzales, Dario Villacreses, Emmett A Dews, Xiaofei Zhou, Ruili Lv, Yuejin Liang.

**Formal analysis:** Ryan H Cho, Lihai Gao, Hui Wang, Yixuan Zhou, Casey Gonzales, Dario Villacreses, Yuejin Liang.

**Funding acquisition:** Lynn Soong, Yuejin Liang.

**Investigation:** Yuejin Liang.

**Methodology:** Ryan H Cho, Xiaofei Zhou, Ruili Lv, Lynn Soong, Yuejin Liang.

**Project administration:** Lynn Soong, Yuejin Liang.

**Resources:** Lynn Soong, Yuejin Liang.

**Software:** Lynn Soong, Yuejin Liang.

**Supervision:** Lynn Soong, Yuejin Liang.

**Validation:** Ryan H Cho, Lihai Gao, Hui Wang, Yixuan Zhou, Xiaofei Zhou, Ruili Lv, Yuejin Liang.

**Visualization:** Lihai Gao, Casey Gonzales, Yuejin Liang.

**Writing – original draft:** Ryan H Cho, Yuejin Liang.

**Writing – review & editing:** Ryan H Cho, Hui Wang, Yixuan Zhou, Hema P Narra, Lynn Soong, Yuejin Liang.

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
