## [Decision Letter · Decision Letter 0]

29 Aug 2025

A Humanized IFN-γ Mouse Model Reveals Skin Eschar Formation, Enhanced Susceptibility and Scrub Typhus Pathogenesis

PLOS Pathogens

Dear Dr. Liang,

Thank you for submitting your manuscript to PLOS Pathogens. Your manuscript was evaluated by members of the editorial board and three external referees. There was a difference of opinion among the reviewers, particularly regarding the validation of the commercially available humanized mouse model. Therefore, we invite you to submit a substantially revised version of the manuscript that addresses all the points raised during the review process and includes a substantial characterization and clear description of the animal model. In addition, Reviewer 1 noted that the appropriate control (Ifngr-/- mice) was not included in several figures; although not all experiments need to be repeated with this control, we strongly encourage revision of selected experiments to enhance the scientific rigor of the study.

Please submit your revised manuscript within 90 days. If you will need more time than this to complete your revisions, please reply to this message or contact the journal office at plospathogens@plos.org. Please include the following items when submitting your revised manuscript:

We look forward to receiving your revised manuscript.

Kind regards,

Joao H.F. Pedra

Guest Editor

PLOS Pathogens

D. Scott Samuels

Section Editor

PLOS Pathogens

Sumita Bhaduri-McIntosh

Editor-in-Chief

PLOS Pathogens

orcid.org/0000-0003-2946-9497

Michael Malim

Editor-in-Chief

PLOS Pathogens

**Reviewers' Comments:**

Reviewer's Responses to Questions

**Part I - Summary**

Reviewer #1: This study by Cho et al examine pathogenesis and disease of Orientia tsutsugamushii in mouse model where human IFNGR1/2 and IFN-g introduced into mice lacking corresponding mouse homologs. The authors examine variety of disease parameters in these mice. Overall, this is an intriguing approach and novel animal models are needed to study scrub typhus. However, the major drawback of this study is that this animal model appears to be provided from a vendor without published validation, and thus it is impossible to decipher what the phenotypes in this study mean. Significantly more validation and characterization of this mouse model should be performed. Such characterizations will be beneficial to the field as well as this model could potentially be used for a variety of approaches even outside of infectious disease. Another major drawback is the lack of the Ifngr-/- control throughout the study, making it unclear if this model is beneficial over the existing model.

Reviewer #2: In “A Humanized IFN-γ Mouse Model Reveals Skin Eschar Formation, Enhanced Susceptibility and Scrub Typhus Pathogenesis”, Cho, et al. addressed a critical gap in the Orientia field with the lack of an appropriate mouse model to study the immunopathogenesis of human disease. The authors circumvented this issue by using a commercially available humanized mouse model that recapitulated human IFN-γ response and eschar formation, along with elevated bacterial burden in disseminated tissues. Although this mouse model requires additional characterization during the course of Ot infection, this study presents a novel model to potentially study adaptive immune responses for vaccine development and disease pathology, such as eschar formation, associated with scrub typhus.

Reviewer #3: In this manuscript, the authors aimed to create a mouse model for studying Orientia tsutsumagushi (O.t.) infection that more accurately reflects what is observed during human infection. The authors note that recent reports suggest an important role for IFN-γ in protection against O.t. Accordingly, they created a genetically modified mouse strain that lacks mouse IFN-γ, and instead has human IFN-γ and IFNGR1/2 receptors knocked in. The humanized mice were more susceptible to O.t. infection as assessed by weight loss, disease severity, the formation of eschar lesions, and histology of skin, liver, lung, and brain. Humanized mice showed elevated inflammatory cytokine responses during infection and disrupted physiological homeostasis as assessed by serum biochemical markers. Higher bacterial burden, reduced interferon-stimulated gene expression, excessive T cell activation, reduced T regulatory responses, and enhanced neutrophil responses were also observed with the humanized mice infected with O.t.

Overall, this is a well-thought-out study that assesses multiple inflammatory and infection parameters to validate a novel humanized mouse model. This is an important technical advancement for this understudied field of vector-borne disease. A few suggestions are noted:

**Part II – Major Issues: Key Experiments Required for Acceptance**

Reviewer #1: The first major problem of this study is lack of validation of the “humanized mouse” that is used. There is a lack of description of it - no description of it in the intro or results, one line in the Methods of where it was purchased (Biocytogen), and in the Discussion the authors state that “this mouse model has never been applied in any research study so far”. There are no cited studies or data regarding how the mouse was constructed or whether the human IFN-g and IFN-g receptor are functional and expressed to the same degree as the native mouse genes. What parts of the human genes are swapped? Is the 5’ and 3’ UTR included? The promoter? There is no description about whether the protein products are functional and modified similarly. Trusting the vendor is not validating – if the authors want to publish using these mice, they should validate that they make similar abundancies of IFN-g and the g-receptor. This will also require significantly more validation, for example showing that the hIFN-gamma receptor binds to the hIFN-gamma made in the mouse and elicits similar downstream cytokine signaling. Regarding post-translational modifications (PTMs) on the humanized version – has this been examined by mass spectrometry? A brief search of the mouse and human IFNGR suggests that they have different PTM sites (https://research.bioinformatics.udel.edu/iptmnet/entry/P15260/ and https://research.bioinformatics.udel.edu/iptmnet/entry/P15261/) . There is no discussion about the regulation of human versus mouse IFNGR / IFN-g – are the genes regulated at the mRNA level? The protein level? Epigenetically?

Figure 3 and Section in lines 320-336 start to address the above point by measuring transcripts of human vs mouse IFNGR, however the data appear to show that human IFNG1 and IFNG2 are only expressed at a 1 “relative fold change” – what does this mean? Wouldn’t a 1-fold change mean no change? The more important comparison to make is to ask how does the mRNA abundance of mIFNGR compare to hIFNGR? If hIFNGR is expressed 2x or even 10x less than the mIFNGR, this could explain the observed phenotypes, assuming that the protein products were functional. In 3C, the most important comparisons are not performed – which would be to compare human IFN-g abundance to mouse IFN-g abundance by Western to determine whether the proteins are expressed to similar degrees in infected and uninfected scenarios, as compared to a control.

The second major problem is that throughout the manuscript, the appropriate control of Ifngr-/- was not used except in Figure 1. If the “humanized” mouse appears phenotypically identical to the IFNGR-/- mouse, is this humanized model a valid mouse model to use or should researchers just use the IFNGR-/- mouse?

For example Figs 4 and 5: The IFNGR-/- control is not included, so it’s unclear whether these phenotypes are due to a dysfunctional hIFNg or just a lack of signaling. Line 377 “the results suggest that human IFN-g signaling confers weaker antibacterial activity compared to its murine counterpart” – based on what specifically? The authors haven’t shown that the hIFN-g is expressed at the same levels as the mouse IFN-g, so it could just be that there is reduced abundance of hIFN-g overall, and it’s slightly upregulated during infection.

Reviewer #2: (No Response)

Reviewer #3: It is possible that human IFNGR 1/2 does not effectively activate mouse JAK-STAT under any condition. This study will be strengthened with the inclusion of a control demonstrating that JAK-STAT signaling is not impaired altogether in the humanized mouse and that the diminished ISG expression reflects a phenotype specific to O.t. infection in this mouse.

Are posttranslational JAK-STAT activation markers also reduced in humanized mice during O.t. infection?

It is unclear why human IFN-γ would cause such significantly different immune responses when compared to mice. The paper would benefit from added discussion on this topic.

**Part III – Minor Issues: Editorial and Data Presentation Modifications**

Reviewer #1: The authors state in the abstract and introduction that “1 million cases occur annually, with more than 1 billion people at risk” and cite Xu et al., 2017 and Luce-Fedrow et al., 2018. Upon inspecting these reports, Xu et al and Luce-Fedrow also make the same assertion, and both papers cite Kelly et al., 2009. However, Kelly et al does not report these numbers. These numbers appear to be frequently reported in the Orientia literature but are in fact not supported by any epidemiology study. In the study “Global seroprevalence of scrub typhus: a systematic review and meta-analysis” by Dasgupta et al., this study itself suggests that the numbers are much lower – although in fact they themselves also repeat the narrative that 1 million people are infected annually, citing Kala et al. However, again, Kala et al do not report these extravagant numbers. The fact that 1 billion people live in areas where scrub typhus occurs does not mean that they are at risk of infection. This would be the equivalent to saying “7 billion people are at risk of tuberculosis infection”. The authors should either provide strong epidemiology studies, which appears to not exist, or should remove these statements.

Line 35- It's confusing to say that animal models are needed to mimic infection via mites, and then the manuscript is not about using mites for infection.

Lines 305-307: The authors suggest that this humanized mouse model is immunocompetent, suggesting that their previous findings with an IFNGR-/- mouse is not immunocompetent? Are IFNGR-/- mice considered immunodeficient as compared to these hIFNGR/hIFNg mice?

Discussion: Since this would be the first description of using this mouse model for an infectious disease study, what other infectious diseases or pathogens would this be useful for studying? A paragraph the potential broad utility of this model would be useful in the discussion for folks in the broader field outside of Orientia.

Throughout the manuscript, referring to these mice as “humanized mice” is confusing and vague. There are hundreds of examples of “humanized mice”. The authors should call them “hIFNGR/hIFN-g mice” in the text and figures.

Reviewer #2: Line 353: As this statement is written, it suggests that WT mice had a significant increase in AMY levels during infection. According to the figure, this is not true when compared to the uninfected control WT mouse.

Figure 1 and 2: It could be due to the quality of the figures, but I am unable to see the scale bar for the skin eschars in Figure 1C and histology images in Figure 2.

Line 401-406: I believe the figure panels are incorrectly cited in the text.

Disease scores for the humanized mouse model appear to mimic that of Ifngr1-/- until day 12 following infection, then gain weight back to comparable levels of WT mice. Most data presented throughout this manuscript pertains to 14 dpi. Could the authors comment as to what immunological shift could be occuring around day 11 to prevent humanized mice from succumbing to infection like Ifngr1-/- mice?

As stated in the introduction, there is extensive antigenic diversity among Ot strains. Could the authors elaborate on information as to whether they believe this humanized mouse model could serve as a model across all Ot isolates?

Reviewer #3: Histology images are dark and have poor contrast. White and color balancing these images will ease interpretation by the reader.

Fig 4A: Graph should be titled IL-1β

Fig 5A: Kidney is not capitalized

Fig 5B: Figure title capitalization should be consistent across panels

PLOS authors have the option to publish the peer review history of their article (what does this mean? ). If published, this will include your full peer review and any attached files.

**Do you want your identity to be public for this peer review?** For information about this choice, including consent withdrawal, please see our Privacy Policy .

Reviewer #1: No

Reviewer #2: No

Reviewer #3: No

**Figure resubmission:**

**Reproducibility:**



---

## [Decision Letter · Decision Letter 1]

29 Jan 2026

PPATHOGENS-D-25-01879R1

A Humanized IFN-γ Mouse Model Reveals Skin Eschar Formation, Enhanced Susceptibility and Scrub Typhus Pathogenesis

PLOS Pathogens

Dear Dr. Liang,

Thank you for submitting your revised manuscript to PLOS Pathogens. Your manuscript was reviewed by the three original reviewers and we all agree that it is much improved. However, there are still some minor issues that need to be addressed as noted by Reviewer 1, including changing the Abstract to better reflect your findings and adding additional statistical analyses.

We look forward to receiving your revised manuscript.

Kind regards,

Joao H.F. Pedra

Guest Editor

PLOS Pathogens

D. Scott Samuels

Section Editor

PLOS Pathogens

Sumita Bhaduri-McIntosh

Editor-in-Chief

PLOS Pathogens

orcid.org/0000-0003-2946-9497

Michael Malim

Editor-in-Chief

PLOS Pathogens

orcid.org/0000-0002-7699-2064

**Journal Requirements:**

**Reviewers' Comments:**

Reviewer's Responses to Questions

**Part I - Summary**

Reviewer #1: Overall the authors have adequately addressed my major points and dramatically improved the manuscript by including better descriptions of the mouse model, additional controls (Ifngr-/- mice), and by addressing the minor points as well. The only two drawbacks of the study is that replacing the mouse IFNGR/IFN-g with the human homologs results in significantly less STAT1 phosphorylation and dramatically less ISG activation, for many ISGs the signaling is more similar to the IFNGR knockout mouse. Thus, the question remains unclear as to whether this mouse has “functional IFN-g signaling” as mentioned in the abstract line 41. The abstract should be accurately reflect the findings – either that this mouse model has aberrant upregulation of ISGs, or something similar, as line 41 would suggest that it is an ideal model. The second minor drawback is that certain statistics should be added, described below.

Reviewer #2: The authors appropriately addressed the reviewer's comments.

Reviewer #3: (No Response)

**Part II – Major Issues: Key Experiments Required for Acceptance**

Reviewer #1: No major experiments are requested to be added.

Reviewer #2: (No Response)

Reviewer #3: (No Response)

**Part III – Minor Issues: Editorial and Data Presentation Modifications**

Reviewer #1: Fig 1B should show the statistical comparisions between Ifngr-/- and hIFNGR/hIFNg at day 11 and the comparison of WT to hIFNGr/hIFN at day 14, even if they are not significant, as these are important to the entire argument about the relevancy of the mouse model.

Similarly, 1D should show the statistical comparison of Ifngr-/- to hIFNGR/hIFNg even if not significant.

It is confusing why Figure 2 the background is yellow and not white? The images look unclear to me. Similar comment – it is hard to discern from these images what the authors are trying to convey, perhaps include arrows pointing out the relevant areas of interest.

Figure 3 and elsewhere – the statistics appear somewhat randomly chosen. Perhaps indicate NS if the comparisons are not significant, or upload a supplemental file with all of the comparisons made to each other and to the controls.

Line 178: should say “were” instead of “was”.

Also line 183 should state what mouse genotypes they are referring to and what control genotypes were included.

Lines 78-80 could mention that compelling epidemiological studies are lacking in the field.

Line 109 should replace “symptoms” with “disease manifestations”

Reviewer #2: (No Response)

Reviewer #3: (No Response)

PLOS authors have the option to publish the peer review history of their article (what does this mean? ). If published, this will include your full peer review and any attached files.

**Do you want your identity to be public for this peer review?** For information about this choice, including consent withdrawal, please see our Privacy Policy .

Reviewer #1: No

Reviewer #2: No

Reviewer #3: No

**Figure resubmission:**
---

## [Editor Report · Decision Letter 2]

2 Feb 2026

Dear Dr. Liang,

We are pleased to inform you that your manuscript 'A Humanized IFN-γ Mouse Model Reveals Skin Eschar Formation, Enhanced Susceptibility and Scrub Typhus Pathogenesis' has been provisionally accepted for publication in PLOS Pathogens.

Best regards,

Joao H.F. Pedra

Guest Editor

PLOS Pathogens

D. Scott Samuels

Section Editor

PLOS Pathogens

Sumita Bhaduri-McIntosh

Editor-in-Chief

PLOS Pathogens

orcid.org/0000-0003-2946-9497

Michael Malim

Editor-in-Chief

PLOS Pathogens

orcid.org/0000-0002-7699-2064

---

## [Editor Report · Acceptance letter]

Dear Dr Liang,

We are delighted to inform you that your manuscript, "A Humanized IFN-γ Mouse Model Reveals Skin Eschar Formation, Enhanced Susceptibility and Scrub Typhus Pathogenesis," has been formally accepted for publication in PLOS Pathogens.

Best regards,

Sumita Bhaduri-McIntosh

Editor-in-Chief

PLOS Pathogens

orcid.org/0000-0003-2946-9497

Michael Malim

Editor-in-Chief

PLOS Pathogens

orcid.org/0000-0002-7699-2064